# Relationship between Job Satisfaction and Workload of Nurses in Adult Inpatient Units

**DOI:** 10.3390/ijerph191811701

**Published:** 2022-09-16

**Authors:** María Fuensanta Hellín Gil, José Antonio Ruiz Hernández, Francisco Javier Ibáñez-López, Ana Myriam Seva Llor, Maria Dolores Roldán Valcárcel, Marzena Mikla, María José López Montesinos

**Affiliations:** 1Faculty of Nursing, University of Murcia, 30100 Murcia, Spain; 2Biomedical Research Institute of Murcia (IMIB), El Palmar, 30120 Murcia, Spain; 3Applied Psychology Service (SEPA), Faculty of Psychology, University of Murcia, 30100 Murcia, Spain; 4Education Faculty, University of Murcia, 30100 Murcia, Spain

**Keywords:** job satisfaction, personal initiative, work environment, staff workload, nursing staff, hospital

## Abstract

Among Nursing and Psychology professionals, the job satisfaction of those in Adult Inpatient Units is analyzed, with a new scale to measure nursing workloads validated. **Objective**: The objective of this study was to relate nursing workloads to professional job satisfaction. **Methods:** This is an observational, analytical, descriptive, concurrent and quantitative study, which used the Overall Job Satisfaction scale and subscales therein, to identify global satisfaction—intrinsic or related to motivational factors and extrinsic or associated with hygienic factors—in nursing professionals (n = 104) from eight Inpatient Units of Internal Medicine and Surgery, in four hospital centers, to describe job satisfaction in the professionals studied and to find statistically significant associations between job satisfaction and workload (measured with the scale MIDENF^®^) in the inpatient units where they work. **Results:** There were higher levels of satisfaction in the variables “relationship with immediate boss” and “relationship with fellow workers”, and lower levels in “relationship with senior management” and “organizational system of the unit”. In the inferential analysis, the scores were 75.63 for overall satisfaction, 35.28 for intrinsic satisfaction, and 40.36 for extrinsic satisfaction. **Conclusions:** There is a close relationship between workload and job satisfaction, showing more dissatisfaction regarding organizational aspects and professional recognition.

## 1. Introduction

Satisfaction has different meanings [1], such as the “Action and effect of satisfying or being satisfied”, “Presumption, vainglory. Having a lot of self-satisfaction”, “Confidence or security of mind” or “Fulfillment of desire or taste”, highlighting in these statements the implications that differentiate the coverage of human needs with the satisfaction we experience when we have those needs covered, depending on actions, responses, feelings, emotions, perceptions, etc. When speaking of “job satisfaction”, we can interpret the above-mentioned definitions, such as Locke’s [2], as “A positive and pleasant emotional state resulting from the subjective perception of the subject’s labor experiences”, understanding that “satisfaction” contains a high degree of subjectivity. Thus, we can deduce that, faced with the same situation, in our case related to nursing care, each professional may feel different levels of satisfaction.

International organizations such as the World Health Organization (WHO), the International Council of Nurses (ICN) and the International Labor Organization (ILO) have identified inadequate working conditions that can affect nurses’ health and job satisfaction, including increased workload, lack of human resources, fixed-term contracts that decrease job security, lack of supplies to provide services and low salaries. These factors generate work overload, fatigue, physical and mental exhaustion, and a high level of stress in this profession [3,4,5,6]. According to the Organization for Economic Cooperation and Development (OECD), the data it offers are considered standard for the countries involved, establishing an average European nurse–patient ratio of 8.8 nurses per 1000 inhabitants. In Spain, on the other hand, this ratio is 5.6 [7]. The poor ratio of nurses in health care centers generates work overload, fatigue, physical and mental exhaustion, and a high level of stress in this profession [5]. In addition, the peculiarities of the type and temporality of employment contracts [8,9] directly influence the satisfaction of health care professionals [10]. There is evidence that a high workload and the failure to match the number of nurses to the actual needs of care lead to understaffing (higher patient–nurse ratios) [11], and leads to an increase in job dissatisfaction, resulting in a higher probability of mistakes, decreased patient safety and reduced quality of care [12,13].

The pandemic revealed that, regardless of the country, health service, health discipline or professional status, not only does excessive workload in the health field generate dissatisfaction problems among professionals, but so does organizational management factors and skills along with workload and low social recognition of health care professionals [14,15,16,17]. In the nursing profession, there are several studies that highlight the high level of labor stress to which nurses are subjected, both psychologically and emotionally [5,14,18,19,20]. This is due to the direct contact with the suffering, pain or death of the patients in their care, as well as the workloads and lack of available resources [18]. According to Maslach’s conceptualization [21], burnout is a response to excessive stress at work, characterized by feelings of emotional exhaustion and lack of emotional resources (emotional exhaustion), a negative and detached response to other people and loss of idealism (depersonalization), and a decrease in feelings of competence and performance at work (reduced personal achievement) [21]. This reality reveals that there is a multi-causality in terms of the factors that generate job dissatisfaction. At the same time, there are several negative effects of this dissatisfaction on the professionals.

Burnout, classified by the WHO as an occupational disease, is a consequence of chronic stress [3,18]. It should be noted that this syndrome has three dimensions: emotional exhaustion, depersonalization and sense of low personal accomplishment [22]. This can cause emotional, behavioral, psychosomatic and social alterations, as well as loss of work efficiency and disturbances in family life [18,23]. Within burnout, Maslach’s theory stands out [21], considering it as a state that occurs as a result of a prolonged mismatch between a person and at least one of the six dimensions of work (workload, control, reward, community, equity and values). Maslach theorized these six characteristics of work as factors that cause burnout and placed the deterioration of employees’ health and work performance as outcomes resulting from burnout [21]. As stated in his theory, he considers workload as one of these dimensions [22,23]. Therefore, workload and staff inadequacy appear as the origin of job dissatisfaction, and burnout, physical and emotional exhaustion, and depersonalization appear as a consequence [11,19,24,25,26]. The relevance of the topic has generated interest, as it has been the subject of research studies on job satisfaction and burnout syndrome in nurses [3,6,10,11,14,17,27], with the aim of making visible the working conditions and needs to which they are subjected.

The phenomenon of job satisfaction was explained by Frederick Herzberg in his Two-Factor Theory [28], which indicates that there are two types of factors that influence job satisfaction. On the one hand, there are hygiene factors (extrinsic or maintenance factors), such as job security, salary, fringe benefits, and working conditions. On the other hand, there are motivators, such as the high qualification required by the job, recognition for better performance, responsibility, autonomy, meaning, involvement in decision making, and organizational commitment. Therefore, job satisfaction among nurses can be influenced by a variety of extrinsic and intrinsic factors.

The job satisfaction of employees has a direct influence on the quality of service, patient satisfaction and organizational development [29], understood as a favorable perspective, with a balance between people’s work expectations, the rewards it offers, interpersonal relationships and the type of management [30,31]. Burnout or job dissatisfaction can have negative effects on the quality of care, patient safety, adverse events, error reporting, infections, patient falls, patient dissatisfaction and family complaints, etc., directly influencing quality of care and patient safety [11,29,32,33]. Job satisfaction also contributes directly to improving performance and results at work, improving productivity at work, and reducing absenteeism and staff turnover among professionals in different services [6,8,9,31,32,34,35]. Thus, job dissatisfaction can have serious consequences for both the professional and the patient.

Several studies [11,36] addressed job satisfaction as a predictor of burnout and concluded that higher levels of job satisfaction are associated with lower levels of burnout and all the dimensions included in Maslach’s theory [21,22,23]. The bibliographic studies consulted present much scientific evidence on the association between workload and job satisfaction, which shows that the “work pressure” in care or management [11,33] has a significant impact, regardless of the sociodemographic variables and working conditions of the professionals. Additionally, the importance of the amount of work and the time to perform it is discussed, including the care and management functions in terms of care activities, defining this association as “quantitative demands” in nursing that result in psychosocial problems among professionals and, hence, affect job satisfaction [11,34]. The characteristics of the jobs that contribute to workload, as well as staffing levels, were the factors most frequently examined in relation to burnout in the studies reviewed, with the authors of [11] finding an association between high workload and burnout. This situation of nurses is a worldwide problem in all specialties but is exacerbated in medical-surgical areas. In this field, high patient–nurse ratios, use of point-of-care technologies, and stressful working conditions require enough highly skilled nurses, with little research available on perceived workload, burnout and intention to quit among medical-surgical nurses [35].

Following this line of research that associates workload with job satisfaction, without going into the obvious negative consequences discussed above, the study we present is one of the phases included in a research project funded by the Carlos III Health Institute [37], related to the State Plan for Scientific and Technical Research and Innovation 2017/2020, aimed at projects and initiatives in health services research, as a research priority in the challenge “Health, demographic change and welfare” [4], and within the “Spanish ERDF Pluri-regional Operational Program (POPE) 2014–2020” (PI18/00950) [38]. The general objective is to contribute to improving the management of nursing human resources, the quality and administration of care, in order to guarantee patient satisfaction and safety. This has been performed through the design, validation and multi-center application of a nursing workload measurement scale for Adult Inpatient Units, based on nursing interventions (Nursing Interventions Classification, NIC) [39,40] validated [41] and registered as MIDENF^®^. In this phase of the project, the main objective was to relate the workloads measured with this instrument to the job satisfaction of the professionals working in the same study units selected for this entire project. We wanted to end our research project with this study on job satisfaction among nurses to visualize the working conditions and needs of the nursing job [3], since knowing both the workload to which they are subjected and their satisfaction with it are the first steps to providing solutions to improve their working conditions and, therefore, the quality of care offered to the population.

## 2. Materials and Methods

This study on satisfaction is part of a project focused on the application of a newly created scale for measuring workloads in nursing professionals, previously validated [41] through a qualitative and quantitative methodology registered as “MIDENF^®^”. In its validation, a Cronbach’s alpha of 0.727 was obtained, considered as acceptable, with a reliability of 0.685. Furthermore, an AVE of 0.099 was obtained, as well as an Omega coefficient of 0.704, a construct validity obtained through a KMO of 0.5 and a significant result in Bartlett’s test. The aforementioned scale contemplates the four nursing functions (care, teaching, management and research). Prior to this study, it was applied in a multi-center manner in the units of Internal Medicine (IM) and Surgery of 4 hospitals, where nursing workloads were measured 2 days a month for 9 months throughout 2020, registering the data in a specific software designed for this scale, and analyzing it with the statistical program R version 4.0.3 (R Foundation: Vienna, Austria) [42].

Starting from this premise and its background, such as Maslach’s theory [21], where one of the six dimensions of work is workload, which is closely related to job satisfaction [11], we hypothesize that a greater workload has a negative influence on job satisfaction and check whether or not this is true in our research project. Among Nursing and Psychology professionals, the job satisfaction of professionals in Adult Inpatient Units is analyzed, in terms of workload and certain sociodemographic variables. As such, this study has the following objectives:To determine the levels of job satisfaction of nursing professionals in the Internal Medicine and Surgical units of four hospitals.To identify overall job satisfaction, intrinsic satisfaction, which is related to job recognition, responsibility, promotion, task content, etc., and extrinsic satisfaction, which is related to aspects of work organization.To determine the association between the different nursing workloads and the levels of overall, extrinsic and intrinsic job satisfaction, by units and hospitals.

### 2.1. Type of Study 

An observational, analytical, descriptive, concurrent and quantitative study was conducted to describe job satisfaction in the professionals studied and to find statistically significant associations between job satisfaction and workload in the inpatient units where they work.

### 2.2. Timing of the Investigation

The data collection phase related to workloads was carried out throughout the year 2020, for 9 months, from January to February and from June to December, since this study had to be stopped during the months of March, April and May due to the COVID-19 pandemic. The data collection phase related to the job satisfaction of the professionals and its analysis was carried out from October 2021 to January 2022.

### 2.3. Sample Analyzed and Scope of Study 

This study was conducted in 8 Adult Inpatient Units, 2 units per hospital, one for Internal Medicine (IM) and one for Surgery for each of the 4 General University Hospitals studied (Table 1). Hospital A is a regional referral hospital, Hospital C is a county referral hospital, and the 4 hospitals are referral hospitals for their respective health areas. The number of rooms per unit ranges from 14 to 20, most of them double occupancy, although there are also some single-occupancy rooms, which represent between 28 and 30 hospital beds per unit.

Of the 8 participating units, 3 units have 13 nurses, 4 units have 14 nurses and 1 unit has 12 nurses. All nurses work 12 h shifts, divided into two shifts: a day shift (from 8 a.m. to 8 p.m.) with 3–4 nurses, and a night shift (from 8 p.m. to 8 a.m.) with 2 nurses (Table 1). In these units, the MIDENF^®^ scale [41] was previously applied, as described above, to determine nursing workload, a scale completed by the same professionals whose job satisfaction has been measured in this study.

This study consisted of applying the Overall Job Satisfaction scale developed by Warr, Cook and Wall [43] in a population of 107 professionals (N = 107) assigned to the aforementioned units of the 4 hospitals under study, collecting a sample of 52 Internal Medicine (IM) nurses and 52 Surgery nurses (n = 104), establishing a compliance rate of 97%.

### 2.4. Evaluation Instruments and Variables Considered

As independent variables, sociodemographic and work-related items were evaluated for the professionals of the sample (sex, age, length of job service in the services/units analyzed and length of working life). These variables were added to the two main variables of this study: job satisfaction and workload of nursing staff in Adult Inpatient Units in the specialties of Internal Medicine and Surgery. The dependent variables were those related to job satisfaction, “overall satisfaction”, “extrinsic satisfaction” and “intrinsic satisfaction” of the professionals, and those related to workload, “overall workload”, “care workload” and “management workload”, obtained in the same Inpatient Units. 

To measure job satisfaction, the Overall Job Satisfaction scale developed by Warr, Cook and Wall [43] was used. This scale assesses job satisfaction, reflecting the experience of workers in a paid job and identifying the affective response to the content of the job itself. It is designed with 15 items and based on Herzberg’s “Two-Factor Theory” [44,45], where both the intrinsic and extrinsic aspects of working conditions are addressed through dependent variables as two subscales:Subscale of intrinsic factors (called “motivational factors”): This subscale includes 7 items (2, 4, 6, 8, 12 and 14) on aspects such as recognition obtained for the work, responsibility, promotion, and aspects related to the content of the task.Extrinsic factors subscale (defined as “hygiene factors”): This subscale includes 8 items (1, 3, 5, 7, 9, 11, 13 and 15) that contemplate job satisfaction, with aspects related to work organization such as working hours, pay, and physical working conditions.

The scale has seven points ranging from “very dissatisfied” to “very satisfied”, resulting in three scores corresponding to “overall satisfaction”, “extrinsic satisfaction” and “intrinsic satisfaction”, obtaining the total score by adding the results given in each of the 15 items by the respondent, assigning a value of 1 to “very dissatisfied” and a value of 7 to “very satisfied”. The total scale score ranges from 15 to 105, where a higher score reflects a higher overall satisfaction. The intrinsic and extrinsic satisfaction subscales were evaluated independently, with values between 7 and 49 for intrinsic satisfaction and between 8 and 56 for extrinsic satisfaction due to their shorter length.

Although this scale does not allow us to establish objective analyses of good or less good working conditions, it does allow us to determine the experiences and opinions expressed by the professionals about these working conditions. Knowing the concepts of intrinsic satisfaction and extrinsic satisfaction enables identifying mechanisms of action aimed at improving job satisfaction related to the content of the job, to give more meaning to it, to provide the worker with greater autonomy, responsibility and control over their own work, to assign more specialized tasks, and to provide the worker with direct information on the results of their work.

To measure the workload, a MIDENF^®^ scale [41] was used. It is structured according to the four functional dimensions of the nursing discipline (teaching, research, management and care). Its items are framed within these nursing functions and were elaborated from a selection of nursing interventions (NIC) [39], adapting them to the tasks or activities derived from the most common interventions in the Adult Inpatient Units of Internal Medicine and Surgery. In addition, each item was assigned a specific execution time after a mapping between the real time measured under current care conditions and the time standardized by NANDA (North American Nursing Diagnosis Association), so that it would be as close as possible to the current reality.

The MIDENF^®^ scale [41] consists of 21 items, each item containing one or more NIC nursing interventions associated with the same time of application. The scale is applied to each patient in each work shift, noting the number of times each intervention/item is performed. The total time spent on that patient is calculated by adding the resulting times for each intervention performed. A nurse’s care workload is calculated by adding the time spent on each of the patients they take care of during that work shift. To this time, the time devoted to unit management, teaching and research, during the same work shift is added in order to determine the total workload of the nurse in the work shift measured.

The MIDENF^®^ scale [41] consists of 15 items for the care function, with their corresponding execution times: self-care (17 min), prevention (2 min), medication (9 min), specimens (5 min), health education (3 min), nutrition (7 min), common invasive procedures (11 min), wounds (9 min), fluid therapy (22 min), device care (13 min), monitoring (2 min), airway (6 min), positioning (4 min), comfort (3 min), and patient and family support (8 min); 4 items for the management function: 3 items for patient-related management of 9 min each (which includes care performed on admission and discharge from the unit) and 1 item for unit management, 21 min); one item for teaching (16 min); and one item for research (20 min). 

In addition, it includes a separate set of items considered complementary, which are activities that are usually performed on occasion in these units, although less frequently than the previous ones, and also have their assigned time: cardiac arrest (35 min), complex administrations (chemotherapy 18 min, blood products 10 min), transfers (60 min), occasional invasive procedures (9 min), isolation (11 min), behavior (50 min), interventions shared with the physician (27 min), and end-of-life care (38 min).

### 2.5. Data Collection Procedure

The data corresponding to workloads were collected during the year 2020, applying the MIDENF^®^ scale [41] to patients admitted to the Internal Medicine and Surgery Inpatient Units of the 4 participating hospitals. Data were collected for 2 days per month on each shift for 9 months (January to February and June to December 2020), interrupted in March, April and May due to the COVID-19 pandemic. Each nurse, who then measured their level of job satisfaction, completed a MIDENF^®^ scale [41] for each patient they took care of during their work shift. These data were introduced in a software designed for this study, where data were registered for the same nurse or reference person responsible for this study in each hospital. This procedure was repeated each day of measurement, registering and analyzing the workload of all the patients admitted in the units of study per shift and day, measuring in minutes the attention given by the nurse to each patient and the rest of the activities carried out according to the 4 nursing functions.

The procedure for collecting data related to job satisfaction from the same professionals who participated in the measurement of workloads was then carried out using the Overall Job Satisfaction sale developed by Warr, Cook and Wall [43]. The scale was sent via e-mail or delivered in paper format in person to the Supervisors of the participating units or the Area Supervisors of Healthcare Quality, who distributed it among the personnel of the selected units. After a few days, the completed scales were collected in paper format in person by the principal investigator of the research project.

### 2.6. Statistical Analysis

In the statistical analysis performed with the R program [42], a descriptive analysis of the 18 variables was performed, by units and hospitals, crossing the questions of the different dimensions according to the sociodemographic variables of the questionnaire. A descriptive analysis, a graphic and the inference were presented. At the same time, an inferential study by satisfaction and workloads was developed, crossing the independent variables, or sociodemographic and occupational variables (sex, age, time working in the services/units), with the results of the dependent variables “overall satisfaction”, “extrinsic satisfaction” and “intrinsic satisfaction” of the professionals, as well as with the variables that provided us with the results of “overall workload”, “care workload” and “management workload”, obtained in the same units in the previous study using the MIDENF^®^ scale [41]. Non-parametric tests were used, since they are the most robust test for ordinal data; the Mann–Whitney U test was used for crosses with two-level factors, and the Kruskal–Wallis K test for crosses with factors of three or more levels. When significant differences were detected, the effect size is shown by using Cohen’s d for two-group crossovers, and eta squared for crossovers of three or more levels.

Although the Overall Job Satisfaction scale developed by Warr, Cook and Wall [43] is already validated, a validation of it was also performed in this study with the data obtained, an analysis of the reliability [46] of this scale, understood as the degree of precision offered by a measurement, with 4 indices: Cronbach’s overall alpha, the Composite Reliability (CR) index, the Average Variance Extracted (AVE) index and McDonald’s Omega index [46]. To verify construct validity, an Exploratory Factor Analysis (EFA) was performed, obtaining the Kaiser–Meyer–Olkin coefficient (KMO), after verifying the correlation matrix of the data, in order to identify variables that were poorly or highly correlated, and Bartlett’s test was performed to rule out the similarity of the matrix with the identity matrix. Finally, a descriptive analysis of the levels of overall job satisfaction, extrinsic satisfaction and intrinsic satisfaction was carried out, by unit and shift, in each of the four hospitals analyzed.

## 3. Results

The results obtained from the 104 questionnaires—52 corresponding to IM nurses and 52 to surgical nurses—from the four participating hospitals analyzed 18 variables. The variables corresponding to the mean age by sex of the respondents and the mean time in years of work in the units evaluated yielded very similar results among the units participating in this study (Table 2). A total of 87 women (83.65%) and 17 men (16.35%) were studied, with a mean age of 39.75 years (ranging from 35 to 46 years), an average working life (years worked since finishing nursing degree) of 15.125 years (ranging from 12 to 19 years), and an average of 11.5 years working in the unit studied (from 8 to 18 years). Since there were similar results in all the participating units, the professional profile of nurses working in this type of unit is very similar: a 39-year-old woman, working for 15 years, of which there are 11 in the units studied (Table 2).

In the inference made between the variables of mean time in years worked and each unit studied, using the Kruskal–Wallis statistical analysis, a quasi-significant *p*-value of 0.05755 was obtained. The rest of the analyses between these sociodemographic variables and workload and job satisfaction did not show statistical significance.

The evaluation of global satisfaction corresponding to each item of the Overall Job Satisfaction scale developed by Warr, Cook and Wall [43] was presented (Table 3), showing the percentages corresponding to the maximum and minimum responses obtained for each item, as well as for each response option (from 1 “very dissatisfied” to 7 “very satisfied”). The mean and median obtained for each item were also calculated (Table 3).

Being ordinal data, among the medians obtained in each item, we highlight those corresponding to item 9 “Relationship between management and workers in your firm” and item 11 “The way your firm is managed”, since they have obtained the lowest medians, with a value of 4, and therefore, the highest percentages of dissatisfaction, 35.6% and 45.2%, respectively (Table 3). The highest satisfaction was obtained in item 3 “Satisfaction with your fellow workers” and item 5 “Satisfaction with your immediate boss”, with overall satisfaction percentages of 96.2% and 94.2%, respectively. Both items with more satisfactory results and the items with less satisfactory results correspond to those included in the extrinsic factors group, i.e., those related to aspects related to work organization.

In the interference analysis, statistically significant differences regarding the hospital were obtained for item 1 (“physical work conditions”), 7 (“rate of pay”) and 14 (“variety of tasks performed in their job”) with a *p* value of *p* = 0.00, item 3 (“satisfaction with coworkers”) with a *p* value = 0.01, item 6 (“the amount of responsibility you are given”) with a *p* value = 0.02, and item 10 (“chance of promotion”) with a *p* value = 0.03. Regarding the unit, statistical significance was only obtained for item 13 (“hours of work”), with a *p* value = 0.03. The overall satisfaction obtained throughout this study was 75.63 (on a 10–105-point scale), with a standard deviation of 14.14 (Table 4). 

When observing the different types of satisfaction obtained, intrinsic satisfaction, which deals with aspects related to the recognition obtained for the work, responsibility, promotion, task content, etc., obtained a mean of 35.28 (the scoring scale ranges from 7 and 49), and a standard deviation of 7.25 (Table 4). Extrinsic satisfaction, related to aspects of work organization, and following a rating scale ranging from 8 to 56 points, resulted in an average of 40.36, with a standard deviation of 7.44 (Table 4).

No significant differences were obtained between the sociodemographic variables with workload and satisfaction, and a statistical quasi-significance was obtained between workload and job satisfaction in relation with the hospital studied, especially when it is in general or care type. If we look at the results by unit and hospital, we see that overall satisfaction is very similar in Internal Medicine, with a mean score of approximately 80 points, scoring 66.15 in Hospital D, with a standard deviation of 16.01. This was also the case in the specialty of Surgery, where all hospitals had a mean satisfaction score of approximately 74–79 points, with Hospital D scoring 68.92, with a standard deviation of 18.54 points (Table 5). This is also true when we differentiate between the two types of satisfaction, as the mean obtained in intrinsic satisfaction scores of approximately 35–37 points for the two types of units studied, with the exception of those corresponding to Hospital D, which range between 30 and 32 points. As for extrinsic satisfaction, the average score is between 39 and 44 points, with the exception of the same hospital, which continues to be the one with the lowest score of approximately 35–36 (Table 5).

Regarding the reliability analysis of the scale, it was studied with four indices. First, an overall Cronbach’s alpha of 0.89 was obtained, which is considered excellent. On the other hand, a Composite Reliability (CR) of 0.89, considered good, and an Average Variance Extracted (AVE) index of 0.37, considered fair, were obtained. Finally, a McDonald’s Omega index of 0.94 was obtained, which is considered excellent [46]. To verify construct validity, an Exploratory Factor Analysis (EFA) was performed, after verifying the correlation matrix of the data, in order to identify variables that were poorly or highly correlated, and Bartlett’s test was performed to rule out the similarity of the matrix with the identity matrix (a significant result was obtained that ruled out such similarity). With the EFA, a Kaiser–Meyer–Olkin (KMO) coefficient of 0.82 was obtained, which was considered very good.

As mentioned above, this study is part of a larger research project in which workload has been measured using the MIDENF^®^ scales [41] in the same units and hospitals where this job satisfaction study has been conducted. Therefore, Table 5 also shows the results obtained regarding the measurement of nursing workloads, in general and differentiating between care and management workloads, in order to verify whether there is a relationship between the workload of nursing professionals and their degree of job satisfaction. When the corresponding inference was made in the statistical analysis, no significant association was obtained between these variables. We note that Hospital D is not only the hospital with the highest level of dissatisfaction, but also the hospital with highest workload, both in general terms and in terms of care, the latter being particularly noteworthy, with results clearly higher than the rest of the hospitals in the two units studied (Table 5).

Regarding the inference analysis, no statistically significant differences for these two variables with respect to the sociodemographic variables. Only two quasi-significant differences were observed between general and care workload of the hospital, with a *p* value = 0.08, which corroborates the differences between these data descriptively (Table 5).

## 4. Discussion

Job satisfaction is an indicator of workers’ well-being and quality of working life, so studying this indicator within health care organizations deserves special treatment, given its direct impact on the quality of the service provided [11,32,33,47,48] and patient safety [11,48,49,50,51,52]. Several studies show the evident relationship between job dissatisfaction with the negative consequences that derive from it, such as burnout and depersonalization [49], and a deficient quality of care [11,48], and patient safety [11,48,49,50,51,52], establishing a significant relationship in which higher levels of burnout are associated with lower patient safety [51,52], which is manifested in fewer notifications of incidents and side effects. The seriousness of the consequences that result from job dissatisfaction for both the patient and the professional highlight the interest and the need for research on this topic. 

The COVID-19 pandemic revealed studies on the areas of opportunity in the health system and the necessity of a greater number of nursing professionals, showing a statistically significant association between the work context of nursing professionals, the type of institution where they work, the work shift and the risk of having been infected with COVID-19, something that affected the satisfaction levels of the professionals. These studies also describe the lack of appreciation for nursing professionals and their lack of participation in decision making [53]. The new worldwide challenge brought to us by the COVID-19 pandemic has been the “alarm bell” that has shown worldwide that it is essential to establish measures that analyze those organizational factors that put the professional’s psychosocial health at risk due to potential stress, burnout, and other elements of physical or emotional exhaustion that generate job dissatisfaction and a decrease in performance and quality of care [54].

Given this situation, job satisfaction is subject to different factors related to our work relationships [10], whether with bosses or co-workers, professional recognition in all aspects, organizational and management climate, work–family balance, training and access to promotion, activities and tasks, etc., factors that are not always equally satisfactory for everyone, even if we are in the same working conditions, due to the subjectivity with which we perceive the coverage of needs [35]. Our research has focused on the association between job satisfaction and workload, without going into all the connotation presented by other research on psychosocial health problems, feelings of burnout of nursing professionals, sleep disturbances, stress, etc. [11,19,25], as factors that generate bio-psychosocial consequences that can affect their levels of satisfaction and, in certain circumstances, associated with other sociodemographic and occupational factors [11,55,56].

Job satisfaction among nurses can be influenced by a variety of extrinsic and intrinsic factors [28,45,57]. The results of our study show a relationship between job satisfaction and workload, with high workloads and low levels of satisfaction, as is predictable and appears in other studies [11,19,25,26]. However, at the same time, there is not always statistical significance according to the type of workload (care or management), and according to the parameters analyzed in the evaluation instrument used, since the degree of “intrinsic satisfaction”, “extrinsic satisfaction” and “overall satisfaction”, related to workloads and type of workloads, is not the same—extrinsic satisfaction, related to aspects of work organization, was higher than intrinsic satisfaction. Other studies agree with our study in showing that, analyzing according to Herzberg’s factors [28,45] (intrinsic and extrinsic), when we relate job satisfaction with workload, the items referring to “relationship with fellow workers” and “relationship with immediate boss” generate more satisfaction, showing that the nurses’ work environment influences job satisfaction [56,57,58]. In addition, those items referring to “relationship with managers” and “management of the firm” [56] also have an influence, because the social cohesion of the superior with the rest of the professionals in their discipline and work commitment were positively and significantly related to job satisfaction [59].

Work environment is a well-known predictor of job satisfaction among nurses as an extrinsic factor, while personal initiative may play a role as an intrapersonal (intrinsic) characteristic [57]. In our study and others consulted [25,26,56,57,58], it is evident that the work environment can contribute to improve personal initiative and job satisfaction, since negative work environments affect burnout through job dissatisfaction [11,36], as shown in the results obtained. A higher workload, associated with a more negative environment, is related to lower job satisfaction, as we can see in the results of Hospital D in the two units studied. It is also reflected in that the items that have shown more satisfaction have been those related to “satisfaction with fellow workers” (96.2%) and “satisfaction with immediate boss” (94.2%), with whom you share the daily workday and directly influence the work environment. This is consistent with the evidence [11] that having supportive factors and positive relationships at work, including relationships with other professionals, hospital management, support from the leader or boss, a positive leadership style, organizational support, and teamwork, could play a protective role against burnout, and influence greater job satisfaction, by having a direct effect on emotional exhaustion and personal fulfillment [11,60]. This situation also appears in Hospital B, which has the greatest management workload and a high care load; but since it is a new hospital, with young staff (with more personal initiative) and a better work environment, job satisfaction is the highest both in general and when evaluating extrinsic and intrinsic factors, which confirms that it is the hospital where the staff have spent the most years working on average in the units studied.

Likewise, within the extrinsic satisfaction factors, research focused on nursing professionals in care tasks has shown that the relationship with managers generates stress and dissatisfaction in the workplace [61]. This has also been reflected in our results, obtaining the lowest satisfaction in the items “relation between management and workers” (35.6%), and the “way your firm is managed” (45.2%). All these are considered as factors of “extrinsic satisfaction” related to the organization, and similarly, other publications conclude that decentralization in management would improve the levels of satisfaction in administrative tasks [62]. This shows the need for hospital management to apply new strategies to improve the working conditions of nurses, related to both extrinsic and intrinsic factors, based on the results obtained in studies as diverse as ours and others consulted [35,59,61,62].

As for intrinsic factors, where we obtained lower satisfaction, with statistically significant differences regarding the hospital only in the items related to the “assigned responsibility” and the “variety of tasks you perform in your job”, we can assume that these two aspects would be the most influential when it comes to continuing in the job, since varying tasks and having more responsibility are challenges for the professional that increase their personal initiative and influence their satisfaction. This is also indicated by other studies [3,35,57], where job demands and intrinsic aspects of the job, including role conflict, autonomy and variety of tasks, are associated with some dimensions of burnout and, therefore, with job dissatisfaction [11]. In this way, the influence of intrinsic factors on the professional is demonstrated in a personal way, generating burnout when dissatisfaction related to them increases, and in a professional way, affecting their work situation.

Regarding the degree of professional satisfaction—extrinsic, intrinsic or overall—associated with IM or Surgery units, our results do not show statistical significance between IM and Surgery units (only in the inference analysis, with respect to the unit, statistical significance was obtained in item 13, related to “working hours”). This indicates that the level of satisfaction may be the same in any of the adult units, regardless of the type of patient, since the difference lies in the type and amount of workload, as well as in the work environment [56,57,58] and the perception of workload and burnout among medical-surgical nurses, which significantly influences the intention to leave the current job [35]. Regarding work conditions related to job dissatisfaction, the shift and work schedule stand out in our study. We see that all the units follow a 12 h shift (one day they work 12 h during the day and the next 12 h at night, resting the following 3 days), which has a positive influence on job satisfaction, since it was the only item that obtained statistical significance, which is related to studies that state that having more than 8 days off per month is associated with less burnout and more job satisfaction [11,62]. On the other hand, shifts longer than 12 h have been associated with more emotional exhaustion [11,63]. We can highlight that one of the aspects that most influences job satisfaction is not related to the type of patient or unit in which one works, but rather depends on the working conditions and the relationship between workload and working hours, the latter being an important determinant of professional satisfaction.

Other studies on the levels of satisfaction associated with leadership or management profiles in IM and surgical units claim that, as in our study, they do not differ from each other, finding these differences between these hospitalization and special emergency units [61,64]. In turn, we found bibliographic documentation that shows that aspects related to the development of competencies, the management of units and the relationship with colleagues are well valued in all units, regardless of the type of patient in them, as obtained in our results. Adding some of these studies, it is necessary for nursing managers to include a new work perspective to address the job satisfaction of nurses, taking into account all the attributes that influence the field of nursing [35,59,61,62,64,65]. It is precisely the implications for nursing management that make it necessary to intervene and create new strategies to improve work and favor contractual conditions. There are several proposals to achieve this, where we highlight those focused on professionals, such as promoting teamwork, developing management and leadership skills in nurses, achieving internal promotion, promoting greater participation in decision making and achieving a better balance of power between administrators of health care institutions and health professionals [3,35,57].

Another aspect to highlight is the relationship between the type of workload and satisfaction, since, in our study, the degree of dissatisfaction is higher with respect to the care workload and lower with respect to the management workload, regardless of the type of unit. This is a result that does not coincide with other studies [11,19,25,26,33,34,66], which present more dissatisfaction related to management workload, arguing that an organizational cultural change is needed based on participation, motivation, commitment and involvement, and to increase support for management workload in nursing [66]. Therefore, it is necessary to develop workload measurement scales, as the one created in our research project [41], that identify the type of workload within all nursing functions (care, education, management and research), as well as the difference between special units, such as critical care units, where there is a long track record in this area [67], and inpatient units, less studied so far [68]. In these units, activities differ greatly, since, in this profession, the workload is not only focused on the care function, but is affected by all the activities, of different types, that the nurse carries out to offer quality care to their patients.

## 5. Conclusions

The levels of job satisfaction of nursing professionals in general and regarding intrinsic and extrinsic factors as well as the relationship between workload were determined in IM and Surgery units of the four hospitals that participated, obtaining the objectives set. Both the items with the most satisfactory and the least satisfactory results correspond to those included in the group of extrinsic factors, related to aspects of job organization.Among the extrinsic factors that influence job satisfaction, the work environment stands out, where a high workload, associated with a negative environment, is related to greater job dissatisfaction among nurses. Another factor that negatively affects nurses, regardless of the department where they work, is the relationship with management and its hospital management model. Satisfaction with co-workers and with the immediate boss are the aspects most highly valued by professionals and which generate the greatest job satisfaction.Job satisfaction, whether overall, extrinsic or intrinsic, is similar in any of the units, whether Internal Medicine or Surgery, with no major differences between the hospitals studied.Intrinsic satisfaction (dealing with aspects related to the recognition obtained for the work, responsibility, promotion, task content, etc.) obtained a lower score (mean of 35.28), and therefore more dissatisfaction among the professionals than extrinsic satisfaction, related to aspects of work organization (with a mean of 40.36). This may indicate that the aspects valued by intrinsic satisfaction are not satisfactorily developed in the Inpatient Units, since the personnel tend to value them negatively because they do not develop satisfactorily in the Hospitalization Units [36]. This may show that the aspects valued by intrinsic satisfaction are not developed in a satisfactory way in the Inpatient Units, since the personnel tend to value them negatively because they cannot perform them in their workplaces.No significant differences were obtained between the sociodemographic variables with workload and job satisfaction, which may be due to the fact that the profile of the nurses working in the study units, regardless of the hospital studied, is very similar, as well as the structural characteristics of the units.As for the objective of determining the association between the different workloads in nursing and overall, extrinsic and intrinsic job satisfaction levels, a statistical quasi-significance was obtained between workload and job satisfaction in relation to the hospital studied, generally and based on type of care.

## Figures and Tables

**Table 1 ijerph-19-11701-t001:** Characteristics of the inpatient units under study.

	Number of Nurses on Staff	Completed Questionnaires	Number of Rooms	Number of Beds	Number of Shifts/Nurses per Day
HOSPITAL A: IM	13	13	15	30	6
HOSPITAL A: SURGERY	13	13	14	28	6
HOSPITAL B: IM	14	14	18	35	5
HOSPITAL B: SURGERY	14	14	17	34	5
HOSPITAL C: IM	13	12	16	30	5
HOSPITAL C: SURGERY	12	12	16	30	5
HOSPITAL D: IM	14	13	20	36	6
HOSPITAL D: SURGERY	14	13	20	36	6
TOTAL	107	104	136	259	44
MEAN	13.375	13	17	32.375	5.5

**Table 2 ijerph-19-11701-t002:** Variables analyzed from the participants.

	Sex		Mean Age (Years)	Mean Working Life (Years)	Mean Working Time in the Unit (Years)
	Women	Men			
HOSPITAL A: IM	11	2	37	13	12
HOSPITAL A: SURGERY	10	3	37	13	10
HOSPITAL B: IM	13	1	41	16	18
HOSPITAL B: SURGERY	10	4	40	16	15
HOSPITAL C: IM	10	2	42	17	8
HOSPITAL C: SURGERY	11	1	40	15	12
HOSPITAL D: IM	12	1	46	19	8
HOSPITAL D: SURGERY	10	3	35	12	9
TOTAL	87	17	318	121	92
MEAN	10.875	2.125	39.75	15.125	11.5

**Table 3 ijerph-19-11701-t003:** Descriptive statistic of the satisfaction scale items.

Items	Min	Max	Significance	Median	%1	%2	%3	%4	%5	%6	%7	SD
1	1	7	4.46	5.00	3.85	14.4	11.5	12.5	22.1	30.8	4.81	1.65
2	1	7	5.23	6.00	0.96	2.88	10.6	9.62	24	37.5	14.4	1.36
3	1	7	6.26	6.00	0.96	0.96	0.96	0.96	3.85	49	43.3	0.97
4	1	7	4.73	5.00	8.65	5.77	7.69	13.5	23.1	28.8	12.5	1.76
5	1	7	6.12	6.00	0.96	0.96	1.92	1.92	11.5	41.3	41.3	1.08
6	1	7	5.49	6.00	1.92	0.96	5.77	9.62	14.4	53.8	13.5	1.25
7	1	7	5.02	6.00	5.77	6.73	6.73	6.73	18.3	46.2	9.62	1.65
8	2	7	5.50	6.00	0.00	2.88	1.92	11.5	24	45.2	14.4	1.11
9	1	7	4.03	4.00	16.3	8.65	10.6	18.3	22.1	14.4	9.62	1.9
10	1	7	4.50	5.00	3.85	9.62	14.4	19.2	19.2	25	8.65	1.62
11	1	7	3.78	4.00	10.6	15.4	19.2	19.2	14.4	18.3	2.88	1.69
12	1	7	4.41	5.00	6.73	9.62	12.5	15.4	24	26	5.77	1.66
13	1	7	5.37	6.00	1.92	7.69	3.85	6.73	15.4	47.1	17.3	1.51
14	2	7	5.41	6.00	0.00	3.85	5.77	9.62	15.4	56.7	8.65	1.2
15	1	7	5.33	6.00	5.77	4.81	3.85	9.62	7.69	49	19.2	1.67
Total	1	7	5.04	5.47	4.55	6.35	7.82	10.96	17.3	37.94	15.6	1.47

**Table 4 ijerph-19-11701-t004:** Comparison of the results obtained for the different types of satisfaction.

	n	Min	P1	Median	Mean	SD	P3	Max
Overall satisfaction	104	44	66.5	75.5	75.63	14.14	88	105
Intrinsic satisfaction	104	16	30	36	35.28	7.25	41	49
Extrinsic satisfaction	104	21	36	41.5	40.36	7.44	46	56

**Table 5 ijerph-19-11701-t005:** Comparison of workload and job satisfaction by unit and hospital.

Unit	Hospital	Job Satisfaction (Mean)			Workload (Mean in Minutes in 24 h)		
		Overall	Extrinsic	Intrinsic	General	Care	Management
INTERNAL MEDICINE	A	78.85	41.92	36.92	7599.94	5225.33	2124.83
	B	81.93	44.14	37.79	13,078.61	7772.83	2959.33
	C	79.75	43.25	36.50	9265.67	6268.72	2279.83
	D	**66.15**	**35.54**	**30.62**	**20,934.11**	**15,712.55**	2926.67
SURGERY	A	74.85	39.85	35.00	7707.11	5369	2216.33
	B	79.29	42.07	37.21	12,526.33	7419.28	2881.5
	C	74.83	39.50	35.33	8263.22	5582.05	1997.83
	D	**68.92**	**36.31**	**32.62**	**15,516.67**	**11,276.55**	2703

## Data Availability

The authors, responsible for the content and results presented in this article, declare: The availability of the data presented in the aforementioned article, being available and without access restrictions due to the demand that may occur, always respecting the regulations governing this access to research, through scientific publications. Likewise, they declare the originality of these data, fruit of a research funded by the Carlos III Health Institute, and can be accessed through this article, through bibliographic references, or through the first author of this article, who, in their opinion, at the same time, is the principal researcher of the Research Project that generated the results presented and the corresponding author.

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
