# Peer review of "Relationship between Job Satisfaction and Workload of Nurses in Adult Inpatient Units"

_ijerph, 2022, doi:10.3390/ijerph191811701_

Round 1

Reviewer 1 Report

Dear Authors.

The health services provision sector has had a particular situation in recent years regarding the workload and the relationship with its satisfaction. Just as it happens in the place under study, similar situations are found in Colombia, due to the scarcity of professionals in the area, which has generated an increase in supply and demand regarding wages.

In this sense, I think they have very good material that allows for a stronger discussion and that leads to conclusions that generate elements for managerial decision-making or contribute to the development of public policies.

In my opinion, they should improve the contribution to knowledge and reiterate that they have good raw material for it.

I look forward to reading the improved and published version to share with my colleagues.  

Author Response

Dear reviewer,
Thank you very much for your review. We have tried to improve all parts of the manuscript, introducing new bibliographic references that complete the introduction and discussion, and improving the exposition of the methodology and results.
You can see all the changes made in the attached manuscript, in red font (please see the attachment).
We hope that they are to your liking and we are at your disposal for any suggestion of change.
Kind regards.

Reviewer 2 Report

Dear authors, congratulations for the magnificent work. but I would need some questions to be clarified, the first is because after such magnificent work, the discussion is so poor, I have reread it three times and consulted literature on Pubmed, and there are many articles that could be used to create a more in-depth discussion. That is why I suggest a rewriting of it, and a discussion with most quality and recent quotes.

Author Response

Dear reviewer,
Thank you very much for your review. We have tried to improve all parts of the manuscript, introducing new bibliographical references that complete the introduction and discussion, just as you have suggested. The presentation of the methodology, results and conclusions has also been improved.
You can see all the changes made in the attached manuscript, in red font (please see the attachment).
We hope that they are to your liking and we are at your disposal for any suggestion of change.
Kind regards.

Reviewer 3 Report

Thank you for inviting me to review the subject manuscript. Please find the below review comments for the authors.

Conclusion of the review: Rejection

This was an interesting empirical replication study in the health industry, specifically focusing on the occupation of nurses. Under the “motivation-hygiene” theoretical framework, the authors proposed that the higher the workload of the nurses, the lower their job satisfaction. Here summarized the major shortcomings of this manuscript:

1.     No hypotheses were formed based on the theoretical framework

2.     No correlational/cross-sectional statistical tests were conducted under the “motivation-hygiene” theoretical framework other than the descriptive statistics

3.     No validation (for example, no factor analyses for face validity; no nomological testing results for construct and discriminant validity) of the created measurement of the Nurse workload scale

4.     Lack of quantitative results to support the discussion and interpretation of the relationship between nurses’ workload and job satisfaction in this specific data collection setting.

5.     No conduct of T-test to conclude the difference in workload and job satisfaction within different hospitals in this dataset. 

Author Response

Dear reviewer, thank you very much for your review. We have tried to improve all parts of the manuscript, introducing new bibliographical references that complete the introduction and discussion. The presentation of the methodology, results and conclusions has also been improved. Please see the attachment, where all changes appear in red font.
Regarding your doubts, we will answer below:

Point 1. No hypotheses were formed based on the theoretical framework

Response 1:  Including the hypothesis in the methodology section of the manuscript (please see the attachment).

Point 2.  No correlational/cross-sectional statistical tests were conducted under the “motivation-hygiene” theoretical framework other than the descriptive statistics.

Response 2: Correlational/cross-sectional statistical tests were not carried out, a descriptive analysis was carried out directly and then an inferential analysis, a correlation study could have been carried out, but an inferential study was sought above all, to find out if there were significant differences between each of the types of satisfaction with the type of unit, hospital, workload, etc., statistically speaking.

Point 3.   No validation (for example, no factor analyses for face validity; no nomological testing results for construct and discriminant validity) of the created measurement of the Nurse workload scale.

Response 3: The scale for measuring nursing workload has been previously validated, with the results of its validation and reliability appearing in the methodology section. Regarding the job satisfaction scale, it is a scale already validated by other authors, and we also provide validation and reliability data obtained with it in this article, which can be consulted in the methodology section of the manuscript.

Point 4. Lack of quantitative results to support the discussion and interpretation of the relationship between nurses’ workload and job satisfaction in this specific data collection setting.

Response 4: We have tried to improve the discussion, providing more data and new bibliographical references, to clarify this point (Please see the attachment).

Point 5: No conduct of T-test to conclude the difference in workload and job satisfaction within different hospitals in this dataset. 

Response 5: No se ha realizado la prueba T porque se han realizado pruebas no paramétricas, como se especifica en el apartado de la metodología. Al tratarse de datos ordinales, los test no paramétricos son  más potentes (Kruskal-Wallis K and the Mann-Whitney U).

We hope we have resolved your doubts and that you like the new changes introduced in the manuscript.

Kind regards.
